# Optimization of the Convective Dose in On-Line Hemodiafiltration: Prospective Interventional Cohort Study—Conducted at Soissons Hospital, France

**DOI:** 10.3390/diseases14010020

**Published:** 2026-01-04

**Authors:** Bedel Lukoki-Beudin, Tchilabalo Kakomkate, Wahiba Ibeghouchene, Céline Carreira, Imene Ouertani, Bruce Shinga Wembulua, Yannick Mayamba Nlandu, Yannick Mompango Engole, Marie-France Mboliasa Ingole, Augustin Luzayadio Longo, Francois Musungayi Kajingulu, Jean Robert Rissassy Makulo, Jean Bonny Nsumbu, Vieux Momeme Mokoli, Nazaire Mangani Nseka, Ernest Kiswaya Sumaili, John Bukasa-Kakamba, Hadrian Hoang-Vu Tran, Audrey Thu, Ayrton Bangolo, Izage Kianifar Aguilar, Simcha Weissman, Janette Mansour, Justine Busanga Bukabau

**Affiliations:** 1Nephrology Department, Soissons General Hospital, 02200 Soissons, France; kakom500@yahoo.fr (T.K.); wahibanephro@gmail.com (W.I.); carreira.cel@gmail.com (C.C.); imene.ouertani@ch-soissons.fr (I.O.); janette.mansour@ch-soissons.fr (J.M.); 2Department of Nephrology, Division of Internal Medicine, University Clinics of Kinshasa, University of Kinshasa, Kinshasa P.O. Box 1333, Democratic Republic of the Congo; yannicknlandu@yahoo.fr (Y.M.N.); yannickengole@yahoo.fr (Y.M.E.); mboliasa@yahoo.fr (M.-F.M.I.); augustin.longo@gmail.com (A.L.L.); francoiskajingulu@gmail.com (F.M.K.); vieux.mokoli@gmail.com (V.M.M.); mnsekan@yahoo.fr (N.M.N.); sumailiernest2015@gmail.com (E.K.S.); justinebuk@yahoo.fr (J.B.B.); 3Faculty of Medicine, University of Goma, Goma P.O. Box 03, Democratic Republic of the Congo; bruliawems@gmail.com; 4Division of Internal Medicine, University Clinics of Kinshasa, University of Kinshasa, Kinshasa P.O. Box 1333, Democratic Republic of the Congo; bonnynsumbu@gmail.com (J.B.N.); johnbukasa73@gmail.com (J.B.-K.); 5Department of Internal Medicine, Palisades Medical Center, Hackensack, NJ 07601, USA; izage.kianifar@gmail.com (I.K.A.); simcha.weissman@hmhn.org (S.W.); 6Department of Medicine, Touro College of Osteopathic Medicine, New York, NY 10940, USA; sphyo@student.touro.edu; 7Department of Hematology and Oncology, John Theurer Cancer, Hackensack University Medical Center, Hackensack, NJ 07601, USA; ayrton.bangolo@hmhn.org

**Keywords:** chronic kidney disease, convective volume, hemodiafiltration, optimization, adequacy, blood flow

## Abstract

**Background and Objectives:** On-line hemodiafiltration (OL-HDF) has been proposed as an alternative to conventional hemodialysis (HD) for patients with end-stage chronic kidney disease (CKD). Randomized controlled trials suggest that OL-HDF may reduce mortality, particularly when the convection volume (CV) exceeds 23 L/1.73 m^2^ per session. However, achieving this target depends on local practices and may be limited to selected populations. The CONVINCE trial reported a 97% success rate using a structured optimization protocol, but its applicability to unselected real-world populations remains uncertain. This study aimed to evaluate the incidence of high CV in OL-HDF among unselected patients managed under routine conditions with a standardized optimization protocol. **Methods and Materials:** This prospective cohort study (May–October 2024) included 67 unselected incident and prevalent patients undergoing HD or HDF in a hospital-based dialysis center. All patients were switched to post-dilution OL-HDF following the CONVINCE optimization protocol, which involved stepwise increases in blood flow, adjustment of filtration fraction, and optimization of session duration. **Results:** The mean age was 68.8 ± 14.9 years; 56.7% were male. Blood flow increased from 283 to 338 mL/min (*p* < 0.001), and the use of dialyzers > 2 m^2^ increased from 36% to 68% (*p* < 0.003). Kt/V improved from 1.22 to 1.6 (*p* < 0.01). CV increased by ~2 L from M1 onward and was sustained through M6, correlating positively with blood flow, session duration, and Kt/V (all *p* < 0.01). **Conclusions:** Stepwise optimization protocol enabled sustained achievement of high CV (23.5 L/session) in 62.3% of patients, improving dialysis adequacy.

## 1. Introduction

According to the World Health Organization (WHO), one in ten adults is affected by kidney disease, and the prevalence of chronic kidney disease (CKD) may reach 17% within the next decade. CKD represents a major public health challenge due to its high morbidity, mortality, and economic burden [1,2]. In 2017, an estimated 697.5 million individuals worldwide were living with CKD at various stages, corresponding to a prevalence of 9.1% [1]. In Europe, data from the ERA and REIN registries provide reliable estimates for renal replacement therapy; however, the overall prevalence of CKD remains uncertain, ranging between 7% and 10% in adults [1,2,3]. In Asia, a 2022 study estimated 434.3 million adults with CKD, most of them in China and India [4,5]. In Africa, data are scarce due to the absence of national registries and limited research funding [5,6,7,8]. A 2018 meta-analysis reported an overall CKD prevalence of 15.8%, with higher rates in sub-Saharan Africa (17.8%) compared to North Africa (6.1%) [7]. In the Democratic Republic of Congo (DRC), a population-based study in Kinshasa found a prevalence of 12.4% among adults [6,9].

In France, the prevalence of end-stage CKD is approximately 1200 patients per million population. Hemodialysis (HD) remains the predominant renal replacement modality, highlighting the interest in optimized strategies such as on-line hemodiafiltration (OL-HDF) [2]. Since 2010, five randomized controlled trials (RCTs) have compared conventional HD with OL-HDF. Four showed no significant effect on mortality, largely due to insufficient convective volumes [10,11,12,13,14,15,16,17,18,19,20]. By contrast, the ESHOL trial demonstrated a 30% reduction in mortality at three years in patients achieving higher convective volumes [14,16,21,22]. A subsequent meta-analysis of individual patient data confirmed the survival benefit associated with achieving a convective volume of at least 23 L/session [12].

Despite ongoing advances in HD, mortality remains high. OL-HDF provides enhanced clearance of medium molecular weight toxins, which may improve patient outcomes [10,11,12]. However, attaining high convective volumes is often challenging, especially in elderly patients or those with limited blood flow. Technical adjustments—such as selecting larger needle sizes or dialyzers with greater membrane surface areas—can help optimize convective volume [18]. The CONVINCE trial demonstrated that target CVs could be achieved in 97% of patients using a structured optimization protocol, although patient selection remains a limitation despite the robustness of the findings. In this context, we conducted the present study in an unselected cohort within a large dialysis center in France, aiming to determine the frequency of high convective volume achievement and its implications for dialysis adequacy and patient outcomes in CKD5D.

## 2. Methods and Materials

### 2.1. Nature, Setting and Period of the Study

This study is a prospective cohort conducted at the hemodialysis center of the Soissons General Hospital, which is part of the public, inter-regional Groupe Hospitalier du Territoire Sud-Axonais des Hauts-de-France (GHT SAPHIR). It was carried out over a six- month period from 10 May 2024 to 10 October 2024.

### 2.2. Study Population and Patient Selection

The study covered all non-selective chronic hemodialysis patients followed at the Centre Hospitalier de Soissons during the study period.

Sampling was exhaustive and patients were recruited consecutively. All patients aged 18 and over with CKD5D on intermittent hemodialysis three times a week for at least three months were included in the study. Severe non-adherence to the frequency and/or duration of hemodialysis treatment (< or =one visit out of the required 3 weekly sessions) and a life expectancy of less than 3 months due to non-renal disease were the exclusion criteria.

### 2.3. Dialysis Equipment

Different dialysis machines (Nikkiso DBB 007 (Tokyo, Japan), Bellco Flexya z 2013 roku and Formula therapy (Mirandola, Italy)), dialysis machines and anticoagulation protocols were used according to the center’s routine. Although all the generators were able to adjust the duration and blood flow rate, none of them were able to directly set a target convective volume or filtration fraction. Each machine used a different parameter, such as the substitution rate or flow. To harmonize the approach, we therefore used the filtration fraction as a reference and designed an algorithm to automatically calculate the equivalent parameter depending on the type of generator used.

### 2.4. Study Parameters of Interest

Data were collected from patients’ computerized medical records using software called EASILY. The variables studied included:

**Epidemiological and clinical data**: age in years, sex, dry weight in kg, height in meters, body mass index (BMI) in kg/m^2^ calculated using the formula: BMI = Weight (kg)/(Height (m))^2^, comorbidities, causative nephropathy, type of renal replacement therapy, length of time on hemodialysis, history of renal transplantation, residual diuresis (quantified over 24 h by the patient using a graduated container with a hermetically sealed cap with a capacity of 3 L), type of vascular approach and AVF flow measured by Doppler ultrasound.

**Biological data**: Albumin, N-terminal pro-B-type natriuretic peptide (NT-pro BNP), C-reactive protein (CRP), haemoglobin level, haematocrit level, ferritin, transferrin saturation coefficient, calcium, phosphorus, parathormone and beta-2-microglobulin were taken before dialysis at each time point (M0, M1, M2, M3 and M6).

**Data from hemodialysis sessions**: actual blood flow, dialysate flow, effective dialysis time, hemodialysis technique, substitution volume, net ultrafiltration, convective volume, filtration fraction, Kt/V, erythropoietin dose, AVF puncture needle size, dialysis anticoagulant and dose, oral anticoagulant, hemodialysis circuit coagulation events, hemodialyzer, membrane type and surface area, generator type, and transmembrane pressure (TMP).

### 2.5. Intervention

A post-dilution HDF optimization protocol inspired by the CONVINCE study (see Figure 1) was implemented to increase the convective volume (substitution volume + net UF). The technical parameters adjusted included session duration (extended to 4 h if possible), blood flow rate (increase in steps of 50 mL/min/session to 400 to 450 mL/min), needle size (change from smaller to larger needles), filtration fraction (total UF flow rate/blood flow rate × [1 − hematocrit]) and choice of membranes. Adjustments were progressive, individualized and tolerance-dependent. The dialysate flow rate and dialysis bath temperature remained unchanged throughout the study, at 500 mL/min and 36.5 °C, respectively. Targeted training of nursing staff ensured consistent application. Therapeutic decisions were made under actual care conditions, in consultation with doctors and nurses.

In the dialysis center where the study was carried out, regular monitoring of the arteriovenous fistula (AVF) was carried out, including measurement every two months of the AVF flow rate and the rate of recirculation. Some generators were also able to estimate this rate in real time. Patients with a recirculation rate of more than 10% and/or an AVF flow rate of less than 600 mL/min, confirmed by Doppler ultrasound, were referred to a vascular surgery consultation for angioplasty prior to their inclusion in the study. This explains why all the patients included in the study had no recirculation or recirculation of less than 5%.

### 2.6. Statistical Analysis

Descriptive statistics were used to summarize the basic demographic and clinical characteristics of the patients. Hemodialysis parameters were analyzed longitudinally by comparing baseline and follow-up values using the Wilcoxon signed ranks test for matched continuous variables, and the McNemar test for matched categorical variables. Clinical and paraclinical variables were compared between participants achieving a convection volume (CV) > 23 mL/kg and those with a CV ≤ 23 mL/kg at 6 months, using Pearson’s chi-square test or Fisher’s exact test for categorical variables and the Wilcoxon signed rank test for continuous variables.

The repeated measures correlation coefficient (rmcorr) was used to assess the associations between VC, blood flow and treatment duration, taking into account intra-subject dependence. A *p* value < 0.05 (two-tailed) was considered statistically significant. All analyses were performed using R software (version 4.3.2).

### 2.7. Ethical Considerations

The study complied with the Declaration of Helsinki (2013 revision, Fortaleza, Brazil) and was approved by the Ethics Committee of the Soissons General Hospital on 13 March 2024. Participation was voluntary, with confidentiality and anonymity ensured. Written or oral consent was obtained; oral consent was accepted when literacy or physical limitations prevented written signature.

## 3. Results

This study included 67 patients at baseline (M0), of whom 47 (70.1%) were already receiving OL-HDF and 20 (29.9%) were not. The 20 non-OL-HDF cases were then progressively switched to OL-HDF, of which initially 8 were on HDF-mixed, 6 were on HFR and 6 on conventional HD. One patient was excluded due to AVF surgery during enrollment. Patients were followed over a six-month period to optimize their convective volume (CV) using the optimization protocol. A total of 53 patients completed the study and 14 patients were censored. Figure 2 summarizes the study population.

### 3.1. General Characteristics of the Study Population

The general characteristics of the population are described in Table 1.

During the study, 67 patients were included, with a sex ratio of 1.53 in favor of men. The mean age was 68.75 ± 14.85 years. High blood pressure followed by diabetes mellitus were the most common comorbidities, respectively, in 86.6% and 58.2% of cases. The most common causative nephropathy was vascular in 53% of cases, followed by diabetic nephropathy in 39% and glomerular nephropathy in 16.7%. Only two patients had autosomal dominant polycystic kidney disease (3% of cases). More than half the patients had residual diuresis ≥ 500 mL/24 h in 55.2% of cases. Biologically, the mean albumin level was 35.7 ± 4.3 g/L, and the mean hematocrit level was 34% in 51.52% of cases. Most of the patients included in the study had a correct phosphocalcic balance, with a median phosphatemia of 1.6 (IQR: 0.7–3) and a median calcemia of 2.3 (IQR: 2.2–2.4). With regard to dialysis treatment, the mean duration of dialysis was 218.9 ± 23.9 min per session, and the mean real blood flow was 284.6 ± 29.8 mL/min. The native AVF was the main vascular approach in 73.1% of cases, with a median ultrasound Doppler flow rate of 1000 mL/min (IQR: 825–1275), and the tunneled central venous catheter was used in 26.9% of cases.

### 3.2. Evolution of OL-HDF Sessions After Optimization Protocol

Optimization of post-dilution convective volumes resulted in a significant improvement in several technical dialysis parameters between M0 and M6 as shown in Table 2. Mean blood flow increased from 284.6 ± 29.8 mL/min to 338.5 ± 34.7 mL/min, with a statistically significant increase (*p* < 0.05). A similar, also significant, increase was observed for ultrafiltration volume and substitution volume. However, although there was a slight increase in filtration fraction, this did not reach statistical significance. In addition, the average duration of sessions remained unchanged overall, with the majority of patients refusing to accept longer treatment times.

In terms of dialytic adequacy, a significant increase in mean spKt/V was observed between M1 and M6 (*p* < 0.05), reaching 1.6 ± 0.2 at M6.

### 3.3. Evolution of Convective Volume

Figure 3 shows the evolution of the mean convective volume at each evaluation point, from the initial period (M0) to M6. At baseline, only 8 of the 47 patients on post-dilution HDF achieved a CV ≥ 23 L/session, with a mean of 18.5 ± 3.8 L/session.

After the optimization protocol was introduced, this proportion rose to 27% (17/62), with a mean of 21.0 ± 4.2 L/session. Progress continued steadily, with 31/61 patients (50.8%) at M2, 27/56 patients (48.2%) at M3 and 33/53 patients (62.3%) at M6. Application of the protocol resulted in a significant increase in mean convective volume, with a gain of around 4.5 L/session between M0 and M6 (*p* < 0.05), reflecting a clear improvement in the convective dose delivered post-dilution.

### 3.4. Evolution of TMP

Figure 4 shows the progression of TMP at 120 min and at the end of the OL-HDF session. TMP increased rapidly throughout the session, reaching a mean of 253 ± 35.6 at 120th minute and 275 ± 37.8 mmHg at the end of the session. They also remained within safety limits.

### 3.5. Patient Characteristics by Convective Volume

As shown in Table 3, comparison of clinical and technical characteristics according to whether or not a high convective volume (CV ≥ 23 L/session) was achieved demonstrated that session duration, body mass index (BMI), and blood flow were significantly associated with reaching this threshold. These associations were observed independently of patient-intrinsic parameters, such as comorbidities or routine biochemical variables. In the high convective volume group (CV ≥ 23 L), mean blood flow was significantly higher (299 ± 38 mL/min) compared with the CV < 23 L group (278 ± 24 mL/min, *p* = 0.049). Similarly, mean BMI was significantly higher in the high CV group (27.3 ± 4.4 kg/m^2^) than in the low CV group (24.7 ± 3.2 kg/m^2^, *p* = 0.028)

Finally, the mean duration of dialysis sessions was significantly longer in the CV ≥ 23 L group (228 ± 19 min) than in the CV < 23 L group (204 ± 23 min, *p* < 0.001).

### 3.6. Factors Associated with Reaching High Convective Volume (M6)

Figure 5 illustrate the correlations between CV and different technical parameters of the dialysis session. A significant positive linear correlation was observed between treatment duration and CV (R^2^ = 0.28; *p* < 0.001) (Figure 5a), as well as between blood flow and CV (R^2^ = 0.41; *p* < 0.001) (Figure 5b). There was also a significant association between puncture needle size and CV (Rm = 0.47; *p* < 0.001), with larger diameter needles (lower G) resulting in higher flow (Figure 5c). On the other hand, no significant correlation was found between Kt/V and the convective volume delivered (R^2^ = 0.071; *p* = 0.057), although the trend observed was close to the statistical significance threshold (Figure 5d).

## 4. Discussion

The aim of this study was to determine the incidence and factors associated with achieving a high convective volume (CV ≥ 23 L) in a hospital-based heavy dialysis center using a CV optimization protocol. At the end of optimization and at M6 later, 17/62 (27%) and 33/53 (62.3%) of patients achieved high convective volume, i.e., on average 21 ± 4.2 and 23.5 L/session, respectively, independently of the patients’ clinical and biochemical characteristics. While session duration and FF remained unchanged, mean blood flow increased from 320.3 ± 36.6 to 338.5 ± 34.7 mL/min. Patients with CV< 23 L/session had a higher percentage of central venous catheters (CVCs), shorter session duration and lower blood flow than patients with CV ≥ 23 L/session.

### 4.1. General Characteristics of the Population

This prospective study included 67 patients, with a mean age of 68.9 years, comparable to that reported by Carrera et al. [20], (69 years) and Maduell et al. [21] in ESHOL (68 years), but higher than that observed in the CONVINCE [17] and Turkish [23] studies, which recruited younger and less comorbid patients.

The predominant comorbidities were hypertension (87.9%) and diabetes (59.1%) at higher rates than those reported by Camiel et al. [24] (hypertension: 70.9%; diabetes: 33.7%) and in the CONTRAST [16], CONVINCE [17], ESHOL [21], FRENCHIE [22] and TURKISH [23] studies (diabetes: 23.8 to 37.3%). The high prevalence of hypertension may be linked to the difficulty in distinguishing between essential hypertension and hypertension secondary to CKD.

As regards the etiology of the nephropathy, vascular forms dominated (53%), followed by diabetic nephropathy (39.4%) and glomerular nephropathy (16.7%). This profile differs from that reported by Canaud et al. (diabetic: 30%, vascular: 18%), reflecting the advanced age and greater cardiovascular risk factors in our cohort.

### 4.2. Changes in Dialysis Parameters After Optimization

Mean blood flow increased from 283.9 mL/min at M0 to 331.1 mL/min at M6, reflecting a significant improvement after implementation of the convective volume optimization protocol. These results are comparable to those of the FRENCHIE Study [22], where a similar mean blood flow was reported. On the other hand, the CONVINCE and ESHOL studies reported significantly higher flow rates, reaching 360 mL/min and 380 mL/min, respectively, reflecting more aggressive strategies in terms of flow rate to favor high convective doses. In contrast, the CONTRAST and TURKISH studies [16,23] reported lower flow rates (300 mL/min and 290 mL/min, respectively), lower than the values achieved at M6 in the present study.

These disparities may be explained by several factors:The prevalence of hemodynamic fragility in the populations studied, particularly in elderly subjects;The variability of practices between centers, in particular the size of needles used, vascular puncture protocols and clinical tolerance thresholds;Vascular access (AVF vs. catheter), which may limit flow rates in certain populations.

The mean duration of sessions remained relatively stable in our cohort, rising from 218.9 ± 23.9 min at M0 to 222.1 ± 19.9 min at M6, a modest increase but insufficient to reach international standards. This figure remains below the 240 min generally observed in reference studies such as CONTRAST, CONVINCE, ESHOL, FRENCHIE and TURKISH [16,17,21,22,23]. There are several reasons for this limit: firstly, the refusal of some patients to extend the duration of treatment; secondly, ethical and clinical concerns relating to the tolerance of long sessions in frail or dependent patients; and finally, the sometimes-restrictive logistical organization of centers, particularly in terms of shift rotation.

It therefore appears that although blood flow has been optimized, limiting the duration of sessions is an obstacle to systematically achieving high convective targets, and highlights the need to tailor HDF strategies to the characteristics of patients, their preferences, and the constraints of the healthcare system.

### 4.3. Evolution of Convective Volume and Factors Associated with Achieving High Convective Volume

In this study, the implementation of a structured optimization protocol resulted in a significant increase in mean convective volume, from 18.5 ± 3.8 L/session at M0 to 23.0 ± 4.2 L/session at M6, with the proportion of patients reaching 62.3% at 6 months. This gradual but continuous increase, observed from M1 (27%) then M2 (50.8%) to M6, testifies to the effectiveness of staged support and the feasibility of high volume in routine clinical practice.

In comparison, these results are close to those of the FRENCHIE study (Morena et al.) [22], which showed that achieving a CV ≥ 23 L was feasible in more than 60% of patients treated in heavy centers, particularly when technical adjustments (blood flow > 300 mL/min, duration ≥ 240 min, membrane ≥ 2 m^2^) were systematized. However, the results of the present study are still much lower than those of the CONVINCE study [17] (97% of patients reached the high convective volume) and several other studies [25,26,27]. This can be explained by the selectivity of patients, particularly in CONVINCE, where patients likely to reach the high CV were chosen.

Among the most decisive technical levers, extending the duration of sessions appears to be central. The gradual increase in duration (from 218.9 to 222.1 min) made a significant contribution to the improvement in convective volume. The closer the sessions are to the 240 min recommended by the major trials (CONVINCE, ESHOL, CONTRAST), the greater the probability of achieving a high CV [21]. However, this strategy remains limited in practice by the tolerance of elderly patients, their compliance, and the organizational constraints of the centers.

Blood flow is a second determining factor. In our study, it increased from 283.9 to 331.1 mL/min between M0 and M6, allowing a significant increase in convective volume. This flow rate is comparable to that observed in the FRENCHIE Study [22] but remains lower than that in the CONVINCE and ESHOL studies (360–380 mL/min), which favored more aggressive approaches. The use of larger gauge needles (G15), which facilitate the achievement of high flow rates, has proved to be essential, in line with the conclusions of Cho et al. [18].

With regard to vascular access, although patients with central venous catheters (CVCs) achieved lower convective volumes, the data in the literature are contradictory. Some post hoc analyses (e.g., CONTRAST) [16] show no significant difference between CVCs and arteriovenous fistulas, suggesting that other parameters (care protocol, technical characteristics) may compensate for this limitation.

Finally, it should be noted that patients achieving a high CV also had a satisfactory Kt/V, and those not achieving this target remained within acceptable efficiency thresholds (Kt/V ≈ 1.21). This highlights that quality HDF relies on a multifactorial approach, combining convective optimization, adequate treatment time and individualized dialytic parameters, as also highlighted in the meta-analysis by Vernooij et al. [17].

In short, this study highlights the feasibility and clinical relevance of a strategy to optimize convective volume, even in elderly or frail patients. Achieving the targets depends above all on fine-tuning the technical parameters and adapting to the individual and structural constraints of the care system.

### 4.4. Purification of β2-Microglobulin

This study did not reveal any statistically significant differences in the evolution of β2-microglobulin, a recognized marker of the purification of medium molecular weight uremic toxins (Table 2). However, a decrease in the mean level was observed, from 29.3 ± 9.4 at the end of the optimization phase to 25.4 ± 8.7 at M6. This downward trend suggests a possible improvement in the purification of medium molecular weight uremic toxins in OL-HDF, although this difference did not reach the significance threshold.

This lack of significance could be explained by the limited sample size and the relatively short follow-up period. Furthermore, it would be interesting to extend the analysis to other medium-weight molecules, such as myoglobin or prolactin, in order to deepen our understanding of the purifying potential of OL-HDF on this type of toxin. Further studies would be needed to explore the full scope of this potential and identify its true clinical value.

### 4.5. Strengths, Limitations and Outlook

This prospective, single-center, non-selective study faithfully reflects daily practice in heavy centers, including all dialysis patients and using various generators and dialyzers. The progressive, individualized approach to convective optimization, with no constraints imposed on patients, confers good ecological validity. The absence of adverse events underlines the feasibility and safety of this strategy.

Limitations include the absence of randomization, monocentricity and a predominantly elderly population, which may restrict generalization. The reduction in the number of patients during follow-up (transfers, transplants) was not significant and therefore had no impact on the final result.

Nevertheless, our results suggest that high convective targets are achievable on a routine basis, including in frail patients, with individualized technical and clinical adaptation. Multicenter studies are needed to confirm these data on a larger scale.

## 5. Conclusions

High-volume OL-HDF (CV ≥ 23 L/session) is feasible in a majority of patients treated in heavy centers, including an unselected, elderly and comorbid population. Thanks to a structured optimization protocol (duration, blood flow rate, filtration fraction) and targeted training of care teams, 62.3% of patients achieved their convective objectives on a long-term basis, with an average volume of 23.5 L/session. The improvement in Kt/V and adherence to the protocol (flow rate and needles) testify to the feasibility and clinical acceptability of the treatment. However, the lengthening of sessions remains a limitation to be overcome in order to maximize the convective benefits.

## Figures and Tables

**Figure 1 diseases-14-00020-f001:**
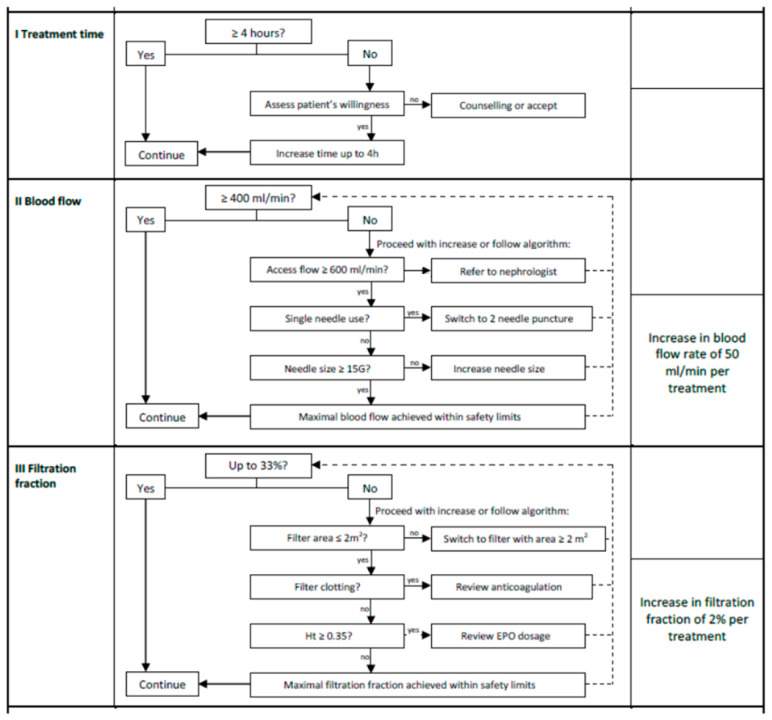
Step-by-step optimization protocol to achieve high convective volume.

**Figure 2 diseases-14-00020-f002:**
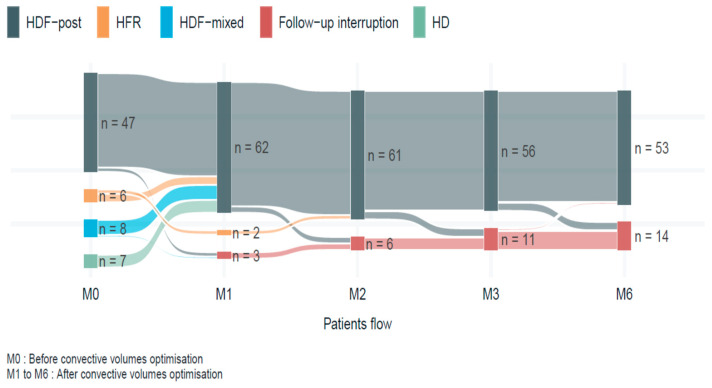
Patient flow.

**Figure 3 diseases-14-00020-f003:**
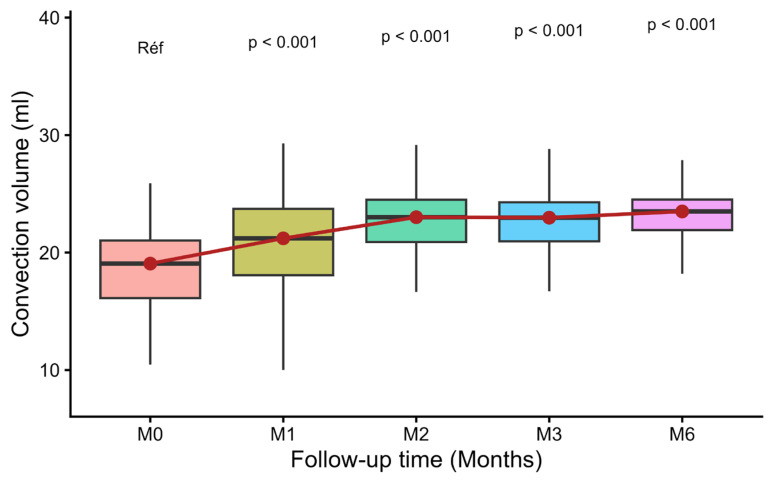
Evolution of mean convective volumes at each time point.

**Figure 4 diseases-14-00020-f004:**
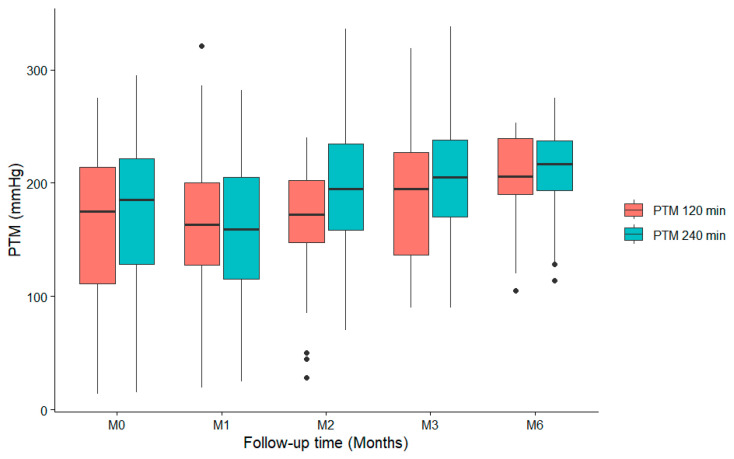
Evolution of TMP from baseline to M6. Footnote bullets represent outliers.

**Figure 5 diseases-14-00020-f005:**
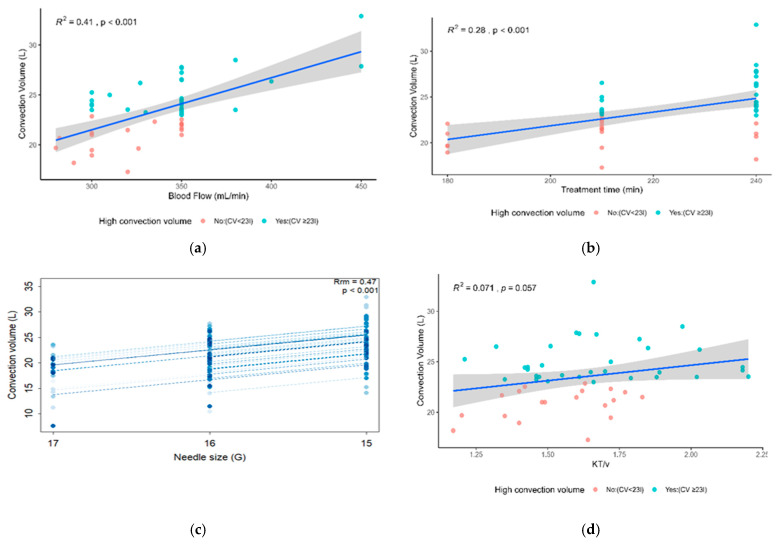
(**a**) correlation between CV and Qblood; (**b**) correlation between CV and session time; (**c**) correlation between needle size and CV; (**d**) correlation between Kt/V and CV.

**Table 1 diseases-14-00020-t001:** General characteristics of the population.

Variables *n* = 67	
***Demographic* ***	
Average age (Years)	68.75 ± 14.85
Men (*n*, %)	38 (56.72%)
Weight (kg)	75.22 ± 16.93
Height (cm)	167.45 ± 10.27
BMI	26.7 ± 4.7
** *Clinical data* **	
Diabetes, *n* (%)	39 (59.09%)
Coronary artery disease, *n* (%)	13 (19.70%)
High blood pressure, *n* (%)	59 (89.39%)
ACFA	16 (24.2%)
Dyslipidemia	48 (71.60%)
Alcohol	25 (37.30%)
Tobacco	35 (52.20%)
Previous kidney transplant	6 (9.10%)
Nephropathy	
*Glomerular nephropathy*	11 (16.67%)
*Vascular*	35 (53.03%)
*Diabetic*	26 (39.39%)
*Polycystic kidney disease*	2 (3.03%)
*CTIN*	8 (12.10%)
*Undetermined*	5 (27.8)
Length of time on dialysis (months)	60 ± 81
History of renal transplantation	6 (9.09%)
Residual urine output	
*<500 (mL/d)*	30 (44.80%)
** *Laboratory data* **	
Albumin (g/dL)	35.7 ± 4.3
Hematocrit	34.9 ± 3.7
Hematocrit > 0.35 (>35%)	34 (51.52%)
Phosphate (mmol/L)	1.6 (0.7–3)
Calcium (mmol/L)	2.3 (2.2–2.4)
NT-proBNP ^¥^	10,802 (2763–11,644)
** *Treatment characteristics (dialysis)* **	
Session duration (min) ^¥^	219 (270–240)
Blood flow (mL/min) ^¥^	286 (270–295)
Vascular access	
*Central venous catheter*	18 (26.9%)
*AVF*	49 (73.1%)
*AVF flow mL/min*	1000 (825–1275)
spKt/V_urea_	1.2 (1.1–1.3)
Net UF (mL)	1761.1 (1075–2215)

* Number (*n*) and percentage (%), mean ± standard deviation; ^¥^: median (IQR), ACFA: atrial fibrillation arrhythmia, CTIN: chronic tubulointerstitial nephropathy, NT-proBNP: N-terminal fragment of B-type natriuretic peptide, AVF: arteriovenous fistula, UF: ultrafiltration, spKt/V: single pool Kt/V, K is the urea clearance of the dialyzer, t the effective duration of the hemodialysis session and V the urea diffusion volume, which is none other than the volume of the patient’s total water (i.e., approximately 60% of post-dialysis weight in men and 55% in women).

**Table 2 diseases-14-00020-t002:** Evolution of OL-HDF sessions after optimization protocol.

Characteristics	M0	M1	*p*	M2	*p*	M3	*p*	M6	*p*
Number of patients	67	62		61		56		53	
Number of patients in OL-HDF	47	62		61		56		53	
Total antico./session (IU)	4484 ± 1860	4644 ± 1789		4535 ± 1868		4509 ± 1793		4592 ± 1914	
Actual blood flow (mL/min) ^§^	284.6 ± 29.8	320.3 ± 36.6	<0.001	332.6 ± 38.7	<0.001	332.8 ± 35.7	<0.001	338.5 ± 34.7	<0.001
Effective session time (min) ^§^	218.9 ± 23.9	224.6 ± 32.7	0.34	222.5 ± 21.6	0.34	222.7 ± 21.1	0.32	222.1 ± 19.9	0.46
Convective volume (CV)	18.5 ± 3.8	21.0 ± 4.2	<0.001	22.7 ± 3.1	<0.001	22.7 ± 3.0	<0.001	23.5 ± 2.8	<0.001
*of which patients with CV > 23 L, n(%)*	8 (17%)	17 (27%)	0.02	31 (50%)	<0.001	27 (48.2%)	<0.001	33 (62.3%)	<0.001
Substitution volume (L) ^§^	16.7 ± 3.7	19.1 ± 3.9	<0.001	20.7 ± 3.0	<0.001	20.7 ± 3.1	<0.001	21.0 ± 2.8	<0.001
Net patient ultrafiltration (mL) ^§^	1777.1 ± 932.5	1927.9 ± 956.2	0.29	2007.5 ± 986.4	0.16	2017.5 ± 1055.2	0.18	2487.7 ± 782.8	<0.001
Filtration fraction (Q Conv/QB) ^§^	29.8 ± 5.0	29.6 ± 5.6	0.27	30.8 ± 3.1	0.43	30.9 ± 3.5	0.49	31.5 ± 2.8	0.13
spKt/V ^§^	1.2 ± 0.2	1.4 ± 0.3	0.017	1.5 ± 0.3	0.035	1.6 ± 0.3	0.01	1.6 ± 0.2	0.01
Beta2 microglobulin (ng/mL) ^§^	27.2 ± 88	29.3 ± 9.4	0.21	-	-	29.5 ± 10.7	0.28	25.4 ± 8.7	0.21
% patients with 15G needle	2 (8.2%)	30 (65.2%)	0.01	26 (59.1%)	0.03	20 (48.8%)	0.03	15 (39.5%)	0.03
Dialyzer ***Adsorbents **** (*n*, %) *******	15 (10.7)	6 (4.4)	0.01	4 (3.0)	0.001	3 (2.5)	0.002	3 (2.7)	<0.001
Membrane surface area (m^2^) *******	1.9 ± 0.2	2.0 ± 0.2	<0.001	2.04 ± 0.1	<0.001	2.1 ± 0.1	<0.001	2.1 ± 0.1	<0.001
of which *n* % surfaces ≥ 2 m^2^	21 (31.3%)	29 (46.0%)	0.06	29 (48.3%)	0.06	25 (43.9%)	0.29	27 (51.9%)	0.15

* Number (*n*) and percentage (%), ^§^: mean ± standard deviation, -: period during which the β-2m assay was performed.

**Table 3 diseases-14-00020-t003:** Distribution of patients according to convective volume at M6.

Characteristics *	CV ≥ 23 L N = 33	CV < 23 L N = 19	*p*-Value
Age (years)	69 (13)	72 (16)	0.5
Sex: M	19 (57.58%)	10 (52.63%)	0.7
BMI (kg/m^2^)	27.3 (4.4)	24.7 (3.2)	0.028
BMI > 30 (kg/m^2^)	21 (63.64%)	7 (36.84%)	0.06
Diabetes	22 (66.67%)	11 (57.89%)	0.5
Coronary artery disease	6 (18.18%)	4 (21.05%)	>0.9
HTA	32 (96.97%)	15 (78.95%)	0.054
ACFA	10 (30.30%)	5 (26.32%)	0.8
Dyslipidemia	24 (72.73%)	11 (57.89%)	0.3
Alcohol	8 (24.24%)	7 (36.84%)	0.3
Tobacco	17 (51.52%)	8 (42.11%)	0.5
atcd_transpl_renal	3 (9.09%)	2 (10.53%)	>0.9
nephro_glomerular	4 (12.12%)	4 (21.05%)	0.4
nephro_vascular	18 (54.55%)	10 (52.63%)	0.9
nephro_diabetic	14 (42.42%)	9 (47.37%)	0.7
CTIN	7 (21.21%)	1 (5.26%)	0.2
old_dialysis_(months)	51 (64)	82 (120)	0.2
Race: Caucasian	32 (96.97%)	19 (100.00%)	>0.9
Black	1 (3.03%)	0 (0.00%)	
Oral anticoagulants	8 (24.24%)	7 (36.84%)	0.3
Residual urine output > 0.5 (L)	16 (53.33%)	8 (50.00%)	0.8
Systolic blood pressure (mmHg)	137(21)	141(18)	0.47
Albumin (g/L)	35.7 (4.4)	34.4 (4.0)	0.2
Hcte > 0.35	20 (60.61%)	10 (52.63%)	0.6
Phosphate (mmol/L)	1.60 (0.50)	1.33 (0.40)	0.067
Ca^++^ (mmol/L)	2.27 (0.17)	2.22 (0.16)	0.2
Ntprobnp (ng/mL)	10,403 (13,134)	13,954 (14,579)	0.4
Session time (min)	228 (19)	204 (23)	<0.001
Blood flow (mL/min)	299 (38)	278 (24)	0.049
Vascular access			
Tunnelled central venous catheter	7 (21.21%)	7 (36.84%)	0.33
AVF	26 (78.79%)	12 (63.16%)	0.2
Kt/V	1.28 (0.22)	1.21 (0.21)	>0.9
Net ultrafiltration (mL)	1850 (931)	1625 (656)	0.3

* Data are expressed as absolute values and percentages, mean ± standard deviation and median with IQR. HTA: arterial hypertension, ACFA: atrial fibrillation arrhythmia, CTIN: chronic tubulointerstitial nephropathy

## Data Availability

The original contributions presented in this study are included in the article. Further inquiries can be directed to the corresponding authors.

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
