# Peer review of "Optimization of the Convective Dose in On-Line Hemodiafiltration: Prospective Interventional Cohort Study—Conducted at Soissons Hospital, France"

_diseases, 2026, doi:10.3390/diseases14010020_

Round 1
Reviewer 1 Report
Comments and Suggestions for Authors
In this study, Lukoki-Beudin and colleagues report a practical investigation aimed at optimizing convective dosing in post-dilution hemodiafiltration and analyzing the most significant influencing factors. For this purpose, they implemented a practice improvement protocol inspired by the CONVINCE trial, applying a stepwise strategy to progressively increase convective volume.
This is a very interesting and clinically relevant study, clearly described and reported, showing that with a simple structured protocol one can markedly improve convective dose delivery in routine practice.
However, I have several concerns that should be addressed by the authors:
- Why were not all patients with AV fistulas ultimately treated with 15-gauge needles, and why was their use limited to only 39% of the population?
- Why was blood flow restricted to an average of 338 mL/min, despite AVF flows of ~1000 mL/min, which would suggest sufficient reserve to increase up to 400 mL/min?
- Why was the filtration fraction limited to ~31%, while in this setting values up to 35% would have been feasible?
- Why was convective volume capped at ~23 L/session (≈90 mL/min convective flow; 80 mL/min substitution flow), while values of 25–28 L/session (≥100 mL/min) could potentially have been reached?
- What was the impact of the dialysis monitors used? Since these HDF machines lack automated ultrafiltration or substitution control, their ability to optimize convection is limited. An analysis stratified by monitor type would be highly informative.
- In Table 2, what is meant by “Dialyzer adsorbent”? Does this refer to an additional adsorptive cartridge placed in the extracorporeal circuit? This should be described and clarified.
- Why was such a high proportion (51%) of large dialyzers (>2.0 m²) used with relatively low blood flows? This mismatch may have reduced efficiency. Dialyzers of 1.6–1.8 m² would likely have been better suited. An analysis by dialyzer type/brand, with clustering by surface area, would be valuable.
- Were patients with residual kidney function included, and if so, was this analyzed as a potential confounding factor on ß2M concentrations?
Author Response
- Why were not all patients with AV fistulas ultimately treated with 15-gauge needles, and why was their use limited to only 39% of the population?
RESPONSE 1 :
At the time of implementing the optimization protocol, our initial objective was to prescribe 15-gauge needles for all patients with an arteriovenous fistula in order to achieve an optimal blood flow rate. However, several factors limited their use. Some patients reported significant pain during cannulation, and prolonged bleeding time was observed in certain cases, despite adjustments in anticoagulation. In addition, the level of familiarity and comfort among nursing staff also played a role, as some nurses were less confident using a larger-gauge needle. Consequently, an individualized approach became necessary, which explains why 15-gauge needles were ultimately used in only 39% of cases.
- Why was blood flow restricted to an average of 338 mL/min, despite AVF flows of ~1000 mL/min, which would suggest sufficient reserve to increase up to 400 mL/min?
RESPONSE 2 :
As described in the “Intervention” section of the manuscript, our initial approach during the M0 phase was to increase the blood flow rate in increments of 50 mL/min per session, aiming for a target of 400–450 mL/min, depending on the patient’s hemodynamic and cardiovascular tolerance. However, this strategy could not be applied uniformly across the cohort. A substantial proportion of patients did not tolerate a 400 mL/min blood flow rate, developing hypotensive episodes and, in some cases, transient myocardial stunning. This limited tolerance can largely be explained by the high cardiovascular-risk profile of our study population, composed mostly of elderly patients with multiple cardiovascular comorbidities, as detailed in Table 1. Therefore, despite the high intrinsic AVF flow, patient safety prevailed, which justified restricting the average blood flow rate to approximately 338 mL/min.
- Why was the filtration fraction limited to ~31%, while in this setting values up to 35% would have been feasible?
RESPONSE 3 :
We remained within the recommended target range for the filtration fraction. Although values up to 35% are theoretically achievable, we observed in our cohort a very rapid and marked rise in transmembrane pressures (TMP) as the filtration fraction approached this threshold. This was likely due to insufficient vascular refilling in this high cardiovascular-risk population, leading to intradialytic hemoconcentration and further accelerating the increase in TMP. In such fragile patients, maintaining the FF at approximately ~31% appeared to be a safer and more reasonable strategy, particularly since this limitation was also related to the restricted mean blood flow rate of 338 mL/min. This pragmatic approach ensured an optimal balance between hemodynamic safety and convective efficiency.
- Why was convective volume capped at ~23 L/session (≈90 mL/min convective flow; 80 mL/min substitution flow), while values of 25–28 L/session (≥100 mL/min) could potentially have been reached?
RESPONSE 4 :
As you know, the primary determinant of the convective volume (CV) is the blood flow rate. In our study, the mean blood flow was limited to 338 mL/min, for the reasons previously explained, which largely explains why the CV plateaued at approximately ~23 L/session. In addition, other convection-related factors contributed to this limitation: notably, 52% of the patients had a hematocrit level > 0.35, reducing the effective plasma volume available for convection and promoting a faster rise in transmembrane pressure (TMP) at higher convective rates. The combination of these hemodynamic and hemorheological constraints made it difficult to achieve convective volumes ≥ 25–28 L/session, despite their theoretical feasibility. Therefore, our approach prioritized a balance between convective efficiency and hemodynamic safety.
- What was the impact of the dialysis monitors used? Since these HDF machines lack automated ultrafiltration or substitution control, their ability to optimize convection is limited. An analysis stratified by monitor type would be highly informative.
RESPONSE 5 : We acknowledge and appreciate this remark. Please see the attachment.
- In Table 2, what is meant by “Dialyzer adsorbent”? Does this refer to an additional adsorptive cartridge placed in the extracorporeal circuit? This should be described and clarified.
RESPONSE 6:
The term “Dialyzer adsorbent” refers to a standard dialysis membrane with enhanced intrinsic adsorptive properties, without the use of an additional extracorporeal adsorptive cartridge. Such membranes combine the conventional mechanisms of toxin removal (diffusion and convection) with adsorption, which enables improved clearance of certain uremic toxins, particularly middle-molecular-weight and protein-bound solutes.
It should be emphasized that this increased adsorptive capacity is associated with specific interactions at the membrane surface. As documented in the literature, these membranes can promote the formation of a protein layer (“protein cake”) on the dialyzer surface, which may increase transmembrane pressure (TMP) and, in some cases, limit the optimization of convection, especially in strategies targeting high convective volumes. This observation is consistent with our high cardiovascular-risk population, in whom a rapid rise in TMP had to be avoided.
- Why was such a high proportion (51%) of large dialyzers (>2.0 m²) used with relatively low blood flows? This mismatch may have reduced efficiency. Dialyzers of 1.6–1.8 m² would likely have been better suited. An analysis by dialyzer type/brand, with clustering by surface area, would be valuable.
RESPONSE 7 :
This is a highly relevant comment. At this stage, I remain uncertain about the most rigorous and appropriate format to present this analysis, and I would appreciate guidance or clarification on the expected methodology.
8. Were patients with residual kidney function included, and if so, was this analyzed as a potential confounding factor on ß2M concentrations?
RESPONSE 8 :
Yes, some patients with residual kidney function (RKF) were included. However, their proportion in our cohort was low, which prevented us from performing a robust statistical analysis to assess RKF as a potential confounding factor influencing ß2-microglobulin levels. The limited presence of RKF may also partly explain the persistently elevated ß2-microglobulin concentrations observed in our population. We acknowledge that in cohorts with a higher prevalence of significant RKF, this parameter should be incorporated as an adjustment variable in analytical models.

Reviewer 2 Report
Comments and Suggestions for Authors
Please, find within the text all suggestions for modifications
The main issue is your Vancouver style in referrence.
References are NOT cited numerically in the text in the order they appear.
In discussion section, try to discuss your results with other similar onot (when needed)

It would be good if you edit the language.
Author Response
Dear Reviewer,
We would like to thank you for your time, your valuable comments, and your constructive suggestions, which have greatly contributed to improving the quality of our manuscript. We would like to inform you that all recommended revisions have been carefully implemented. In addition, some of the responses to the reviewer’s comments can be found directly in the PDF version of the manuscript, highlighted in track changes. Please do not hesitate to contact us if further clarification is needed.
With kind regards,
Dr Bedel LUKOKI-BEUDIN
The Authors

Round 2
Reviewer 2 Report
Comments and Suggestions for Authors
No comments